# Visualization and Quantification of the Penetration Behavior of Bentonite Suspensions into the Pore Network of non-cohesive Media by using μ-CT Imaging

- 5 Britta Schoesser<sup>(1)</sup>, Atefeh Ghorbanpour<sup>(2,1)</sup>, Matthias Halisch<sup>(3)</sup>, Markus Thewes<sup>(1)</sup>
- 6

4

- (1) Institute for Tunnelling, Pipeline Construction and Construction Management, Ruhr Universität Bochum, Universitätsstraße 150, D-44780 Bochum
- 9 (2) LV Baumanagement AG, Banksstraße 4, D-20097 Hamburg
- (3) Leibniz Institute for Applied Geophysics (LIAG), Dept. 5 Petrophysics & Borehole
   Geophysics, Stilleweg 2, D-30655 Hannover

12

#### 13 Abstract

Bentonite suspensions are an essential tool for different construction techniques in horizontal 14 15 and vertical drilling, in diaphragm and bored pile walls as well as in pipe jacking and tunneling. One of the main tasks of the suspension is to prevent the surrounding ground from 16 17 collapsing during the excavation process of trenches, drill holes or tunnels. In order to 18 maintain the soil stability close to the excavation, the bentonite suspension has to counteract 19 against the earth and water pressure. Therefore, the pressure acting in the suspension has 20 to counter the groundwater pressure and to be transferred into an effective stress to support the soil skeleton. 21

The creation of a pressure transfer mechanism can be achieved in two ways. A direct relation exists between the mechanism of the pressure transfer and the penetration behavior of the bentonite suspension in the subsoil. The relation of the size of the bentonite particles in the suspension and the size of the pores in soft soil is decisive. In addition, the yield strength of the bentonite suspension is a determining factor.

Concerning the penetration behavior two theoretical models exist actually: formation of a filter cake and entire penetration into the pore space. If the pore space is smaller than the size of the bentonite particles, a filtration process takes place. Here, the bentonite particles agglomerate gradually at the entrance of the pore space and create a thin nearly impermeable layer. This membrane is named filter cake. If the pore space is larger than the size of the bentonite particles, the suspension penetrates into the subsoil up to a certain depth.

These models have a more theoretical character due to missing visual evidence concerning the interaction of the bentonite suspension in the pore space. Here, the micro CT technique delivers a valuable contribution to this research.

37

<u>Keywords:</u> μ-CT, non-cohesive porous media, bentonite suspensions, non-Newtonian fluid,
 penetration depth, filter cake

#### 40 **1** Introduction

Bentonite suspensions are an essential tool for different construction techniques in horizontal and vertical drilling, in diaphragm and bored pile walls as well as in pipe jacking and 42 43 tunneling. One of the main tasks of the suspension is to prevent the surrounding ground from collapsing during the excavation process of trenches, drill holes or tunnels. In order to 44 45 maintain the soil stability close to the excavation, the bentonite suspension has to counteract 46 against the earth and water pressure. Therefore, the pressure acting in the suspension has 47 to counteract the groundwater pressure and has to be transferred into effective stress to 48 support the soil skeleton.

Currently, the dominant theory in tunneling practice is adopted from diaphragm wall technology [Müller-Kirchenbauer, 1977] and summarized in DIN 4126 (2004). According to German Standard [DIN 4126, 2004] the formation of a pressure transfer mechanism in terms of effective stress can be achieved in two ways: (a) with a thin and flexible membrane or (b) with a limited penetration zone in the soil.

The membrane, named filter cake (a), develops when the pore size of the soil is smaller than the size of the suspended bentonite particles [Walz, 2001]. Here, the bentonite particles are filtered at the entrance of the pore space and the remaining filtrate water drains through the soil. By gradual agglomeration of more bentonite particles, a thin, impermeable membrane is build (Figure 1 (left)). Here, the suspension pressure is transferred through the membrane area in terms of effective stress to the soil skeleton.

In case the pore size of the soil exceeds the size of the suspended bentonite particles, the suspension penetrates completely into the pore space of the ground up to a certain depth [Walz, 2001]. Due to the yield point of the bentonite suspension, shear stresses are transferred along the surface of the soil particles within the penetration zone (b). The penetration process stagnates in a certain depth, when the suspension pressure, transfer of shear stress and groundwater pressure are balanced (Figure 2 (left)).

[Mueller-Kirchenbauer 1977] describes the direct relation between the mechanism of the 67 pressure transfer and the penetration behavior of the bentonite suspension in the subsoil in 68 reference to the pore size in detail.

Figure 1: Theoretical principle of support pressure transfer in the soil due to formation of a filter cake
 [Zizka & Thewes, 2015] (left) and experimental result of filter cake on macroscale [Imerys, 1998]

(right).

Figure 2: Theoretical principle of support pressure transfer in the soil due to formation of a penetration
 zone [Zizka & Thewes, 2016] (left) and experimental result of a penetration zone on macroscale
 [Imerys, 1998] (right).

Based on long-term practical experience, the theoretical principles are widely accepted
[Anagnostou & Kovári 1994, Krause 1987, Boere 2001, Bezuijen 2001, Zizka & Thewes
2015] and in addition proven by several experimental expertise on the macroscale [IBECO,
Min et.at. 2013, Heinz 2006, Arwanitaki 2009] (Figure 1 (right), Figure 2 (right)).

In this study the penetration process is visualized and analyzed on the microscale using  $\mu$ -CT for the first time. The analysis provides detailed information concerning the interaction 83 84 between the bentonite suspension and the non-cohesive media within the pore space. 85 Furthermore, the µ-CT imaging provides the determination of parameters e.g. porosity, pore 86 size, pore size distribution and particle size distribution. By analyzing the contact angle 87 between the fluid and the glass beads, the bentonite suspension is identified as a "wetting 88 fluid" with a contact angle <90°. Both principles - filter cake and penetration - are identified 89 and analyzed using µ-CT imaging so that these phenomena are validated on the microscale.

As an important result, the single phenomena of the filtration process of the bentonite suspension can be demonstrated in detail. Beside the "standard" identification of solid particles (glass beads), air and the penetrated bentonite suspension, the filtered bentonite particles in the filter cake and the filtrated suspension water are detected.

#### 95 2 Materials & Methods

#### 96 2.1 Materials

Bentonite is a natural clay. Main component of bentonite is the plate like clay mineral
Montmorillonite. A single Montmorillonite crystal consists of 15-20 elementary layers.
Between these elementary layers different cations (e.g. Na<sup>+</sup>, Ca<sup>2+</sup>, Mg<sup>2+</sup>) are adsorbed. In
case of Na<sup>+</sup> ions the bentonite is called Sodium bentonite, in case of Ca<sup>2+</sup> ions it is a Calcium
bentonite.

Preparation of a bentonite suspension consists of three steps: The powdery bentonite is suspended in water (1) and dispersed by introducing high shear forces (2). The shear forces separate the single crystal layer mechanically and distribute them homogeneously in the suspension. Due to an additional swelling process (3), water molecules are embedded between the elementary layers of the Montmorillonite crystal. These water molecules are adsorbed at the cations and at the surface of the single layers as well. Hence, the distance

between the layers increases and the volume of the dispersed/suspended solids changes.
This break-up of the layer corpuses is essential for the rheological properties of the bentonite
suspension to develop. The required swelling time of different bentonites varies between 4 16 hours. Afterwards the particle size of the suspended Na- and Ca-bentonite particles can
be determined (Appendix 1). In the experimental study bentonite suspension with varying
solid contents were employed: Ca-bentonite in 25 % by weight, Na-bentonite in 8 % and 13
% by weight.

Glass beads with particle size of 2 mm and 600 µm and a mean density of 2600 kg/m<sup>3</sup> were used to ensure the reproducibility of the performed combinations of bentonite suspensions

and non-cohesive media. The surface structure of the glass beads was determined using

SEM (Figure 3). Here, small parts of unevenness were detected.

Figure 3: Image of the surface condition of the 2 mm glass beads using Scanning Electron
 Microscopy (SEM)

The penetration tests were conducted in test tubes made of acrylic glass (Ø 37 mm and 123 length 160 mm) and silica glass (Ø 21 mm and length 200 mm) in order to provide the most 124 suitable material for the  $\mu$ -CT scans. The label of each sample describes the container type, 125 the type and concentration of bentonite suspension and the size of the glass beads. Table 1 126 shows the combinations for scanning with  $\mu$ -CT, which provide the performance of a filter 127 cake or the penetration process.

| 128 | Table 1: Combinations of test tube material, | bentonite suspensions | s and glass beads size for | CT scans |
|-----|----------------------------------------------|-----------------------|----------------------------|----------|
|-----|----------------------------------------------|-----------------------|----------------------------|----------|

|          | Material of   | Bentonite type + | Size of glass |
|----------|---------------|------------------|---------------|
|          | test tube     | solid content    | beads         |
| Sample 1 | silica glass  | Calcium 25 %     | 2 mm          |
| Sample 2 | silica glass  | Calcium 25 %     | 600 µm        |
| Sample 3 | silica glass  | Calcium 25 %     | 2 mm + 600 μm |
| Sample 4 | silica glass  | Sodium 8 %       | 600 µm        |
| Sample 5 | acrylic glass | Calcium 25 %     | 2 mm          |
| Sample 6 | silica glass  | Sodium 13%       | 2 mm + 600 μm |

129