# Peer review of "Visualization and Quantification of the Penetration Behavior of Bentonite Suspensions into the Pore Network of non-cohesive Media by using μ-CT Imaging"

_Solid Earth, 2016_

## Referee Comment (RC1) · Anonymous Referee #1 · 13 Mar 2016

The authors provide an overview of their experiments about bentonite infiltration into glass beads. The subject fits well to scope of the special issue, as it combines 3D imaging with a geotechnical application. However, I cannot recommend its publication for several reasons.

1. Objective: There seems to be no clear research question, which this study addresses. The three-dimensional visualization of a well studied process with $\mu$-CT should not be the only motivation nowadays. The authors themselves state that from their observations no general rules can be derived about how properties of the sus-

pension and the pore space affect penetration depth (l 472-476).

2. Design of the experiment: - A lot of results were presented that only distract from the central message of the paper without providing further insights. For instance, the material of the cylinder (acrylic vs. quartz glass) had no influence on penetration. This can be stated in one sentence without showing results for sample 5 (Fig. 4, etc.) - Various parameters were changed at the same time, which mixes up the causes for observed effects. I assume Ca and Na changes the flocculation behavior of bentonite (is this stated somewhere in the introduction?), which is why the particle sizes for Ca-Bentonite and Na-Bentonite were different. You state that the viscosity of the suspension was different. Is this due to the different particle sizes (Appendix 1) or due to the different densities (Ca-Bentonite 25%, Na-Bentonite 8-13%)? Which of these effects is governing penetration depth? - Parts of the analysis is of no obvious use for the conclusions. For instance, the contact angle measurements are only shown for single interfaces from three out of six samples (Ca - Bentonite only). There are several issues with that: 1. The authors don't state how the contact angles were measured. The plane in which the angle can be measured without bias, needs to be perpendicular to the three-phase contact line (Andrew et. al. 2014) and many angles need to be measured in order to get robust results. 2. Why would you expect different contact angles, when all material properties are the same (same suspension, beads of same material - only differ in size). 3. How exactly would a change in contact angle effect penetration depth anyway?

3. Style: - Abstract reads like an introduction and not like a concise summary. In fact, parts of the abstract are directly copied from the introduction. The abstract should tell what you did, why you did it and why this is important. - Some figures are in German (Appendix 1). Some are unnecessary, because they only show common knowledge (Fig. 16). Some show conceptional sketches that are not well explained (Fig. 1,2). Some show what seems to be colorful watershed partitioning of the pore space (e.g. Fig.6,14) without telling why this is necessary, because results are only analyzed with

respect to depth profiles anyway. Some figures are of low quality so that observations made in the text are not really supported by the figures (e.g. water films in Fig. 12). - Some of the cited literature is in preparation (Zizka & Theiwes 2015), some is not even listed (Zizka & Theiwes 2016). - Many typos and bad wording. Maybe consider to consult a native speaker. - Some passages of the text are unimportant and should be omitted, e.g. l215-224 or l489-495.

4. Other: - Is the pore size distribution determined by the maximum inscribes sphere method, by the watershed partitioning in individual pores or another method? - a vertical test line through the image is called a profile and not a histogram (which is the frequency distribution of intensities for an entire image). - What does the "measurement" module in Avizo do exactly?

References: Andrew, M., Bijeljic, B., Blunt, M.J.: Pore-scale contact angle measurements at reservoir conditions using X-ray microtomography, Advances in Water Resources 68(0), 24–31, 2014

---

## Author Comment (AC1) · 19 Apr 2016

Dear reviewer, thank you very much for your constructive review. We have summarized all of our comments on your points within the attached .pdf file.

We think that with these changes, we can improve the quality and consistency of our paper.

With kind regards M. Halisch (on behalf of all authors)

Please also note the supplement to this comment:

[Figure]

http://www.solid-earth-discuss.net/se-2016-42/se-2016-42-AC1-supplement.pdf

[Figure]

**Supplement:**

**Author's comments according to anonymous reviewer #1**

The authors would like to thank the reviewer for the constructive review of the manuscript. We will try to address every of your points in the following comment sections.

**Referring to point no.1, objective: clear research question**

We agree with your comment, that the research question needs specification. In reference to point no.2 (design of experiments) we will strengthen the explanation.

**Referring to point no.2, design of experiments**

The tested bentonite suspensions vary with regard to the raw material (Ca- and Na-bentonite) and the solid content. These provide diverse rheological behavior under aspect of flow and filtration tendency in terms of separation of the solid and fluid phase of the suspension within or at the entrance of the porous medium. We agree with your comment that the influence of the parameters to the penetration behavior need a more detailed explanation.

The water absorbency and the swelling rate of natural bentonites differ greatly in respect to the embedded cations Ca or Na. In general, Na-bentonite shows a higher swelling rate than Ca-bentonite with consequences for the experimental layout: Ca-bentonites provide the possibility to generate high values of solid content. For a comparable rheology of the suspension, Na-bentonites require lower solid contents. The separation effect of bentonite particles and water out of the suspension arises in all tests using Na-bentonite suspensions. Due to the lower solid content and slightly larger particle size of the particles, the Na-suspension creates the filtering process at the entrance of the pore space of the porous media and performs the thick layer of bentonite particles (filter cake). As a result, the penetration depth of the Na-suspension is very small. In contrast, Ca-suspension create a larger penetration depth due to the smaller particles size. It is evident for all experiments, that the separation process of solid and fluid phase requires a certain penetration depth and inner shape of the pore structure. We will revise this part carefully.

**Referring to point no.2, contact angle measurements**

We agree with your comment, that this chapter is some kind of "stand alone" chapter, without being directly implemented to the analysis. The idea behind these measurements was to investigate the wetting behavior of the different types of suspensions depending on the formation of a water film (caused by "filtration" of the suspension during imbibition) in front of the suspension. At the time of submission, this part was still under investigation. Unfortunately, the staff working on this has changed its affiliation and working area, hence we cannot improve this part extensively and as needed. We would like to remove this chapter from the manuscript for an overall increased consistency.

**Referring to point no.3, style**

We will revise the abstract to be more a concise summary of the experiments and results.

**Referring to point no.3, colored figures like fig.6, 14, etc…**

Well, these images have been implemented just o give an impression about the overall pore space as well as of the individual "pores". Of course, they are not of utmost importance and can be easily shifted into the Appendix of they can be added as supplementary material. The partitioning of the pore space has been performed to assess the "quality" of the representativeness of the investigated "CT-volume" referred to the experimental volume (table 3 and the related part of the text). We can rephrase this part to be more distinctive.

**Referring to point no.4, Pore Size Distribution**

- The pore size distribution has been determined by watershed partitioning of the pore network. We will add this information.
- For clarification: a histogram is a graphical representation of a frequency distribution of a metrical scaled quantity. Hence, the profile is a 2D or so called line histogram of the frequency distribution of grayscales along the individual profiles. Accordingly, for the entire image this would be a 3D or volume histogram. Hence, we will keep using "histogram" as notation.
- The measurement module is a numerical "toolbox", which features different algorithms for image quantification. Since this knowledge is not essential for the manuscript, we will remove these phrases consequently from the document.

---

## Referee Comment (RC2) · Anonymous Referee #2 · 16 May 2016

This research paper deals with the uCT imaging of transport of Bentonite suspension in glass beads packing of two different sizes (2mm and 600um diameters). The topic is of importance to many applications such as civil engineering and slope stabilization, well clogging, etc. Sodium-Bentonite and Calcium-Bentonite as the two major suspensions have been injected the methodology and objectives are valid and very relevant to porous media applications.

Authors have used uCT imaging to visualize the penetration/filtration of bentonite suspension in glass bead packing. uCT imaging has been used extensively in the

past decade for diverse porous media applications such as particle deposition, flow, transport, multiphase flow. e.g.: uCT imaging of particle deposition in porous media (GRL,34,2007, L18404), MRI imaging of particle deposition (Environ. Sci. Technol. 2005, 39, 7208-7216). Although this approach has been used in these disciplines, the authors introduce the technology for other applications such as tunnel engineering.

Conclusions made are valid within the very limited parameterization space. There are two major drawbacks in this study.

a. In practice, for creating bentonite suspension, water with a given composition and pH is used. Since the swelling behaviour of bentonite and its agglomeration are highly controlled by pH and salinity of the suspension, the authors should have presented a sensitivity analysis of the penetration/filtration process as a function of water ionic strength/pH or at least report their own experimental conditions. Surprisingly, there is no information about the water and the resulted suspension specifications.

b. The porous media is spherical glass beads while natural granular porous media are made of irregular grains which are covered by some impurities such as clays. It is known that movement of particles including bentonite suspension is highly influences by the shape of grains, roughness, and surface properties. Roughness and grain shapes control the pore scale hydrodynamics while surface properties control the surface forces and interaction between the particles and porous media. None of these factors are present in the current study. As the most simple approach, at least crushed glass beads or a simple sand packing could be used to investigate the penetration/filtration processes.

1. Does the paper address relevant scientific questions within the scope of SE? This research paper deals with the uCT imaging of transport of Bentonite suspension in glass beads packing of two different sizes (2mm and 600um diameters). The topic is of importance to many applications such as civil engineering and slope stabilization, well clogging, etc. Sodium-Bentonite and Calcium-Bentonite as the two major suspensions have been injected the methodology and objectives are valid and very relevant to porous media applications.

2. Does the paper present novel concepts, ideas, tools, or data? Authors have used uCT imaging to visualize the penetration/filtration of bentonite suspension in glass bead packing. uCT imaging has been used extensively in the past decade for diverse porous media applications such as particle deposition, flow, transport, multiphase flow. e.g.: uCT imaging of particle deposition in porous media (GRL,34,2007, L18404), MRI imaging of particle deposition (Environ. Sci. Technol. 2005, 39, 7208-7216). Although this approach has been used in these disciplines, the authors introduce the technology for other applications such as tunnel engineering.

3. Are substantial conclusions reached? Since the system of porous media is very simple and the parametric space adopted is very limited (no analysis on the effect of water composition, pH, grain size distribution of the porous media, roughness, natural porous media) the conclusions made dot not provide a substantial contribution to the current understanding.

4. Are the scientific methods and assumptions valid and clearly outlined? uCT imaging technique used in valid and appropriate for this study. However the image processing is very brief. It is needed to mention how the different steps are done. Maybe presenting the image processing in an appendix would be better.

5. Are the results sufficient to support the interpretations and conclusions? Conclusions made are valid within the very limited parameterization space. There are two major drawbacks in this study.

a. In practice, for creating bentonite suspension, water with a given composition and pH is used. Since the swelling behaviour of bentonite and its agglomeration are highly controlled by pH and salinity of the suspension, the authors should have presented a sensitivity analysis of the penetration/filtration process as a function of water ionic strength/pH or at least report their own experimental conditions. Surprisingly, there is

no information about the water and the resulted suspension specifications.

b. The porous media is spherical glass beads while natural granular porous media are made of irregular grains which are covered by some impurities such as clays. It is known that movement of particles including bentonite suspension is highly influences by the shape of grains, roughness, and surface properties. Roughness and grain shapes control the pore scale hydrodynamics while surface properties control the surface forces and interaction between the particles and porous media. None of these factors are present in the current study. As the most simple approach, at least crushed glass beads or a simple sand packing could be used to investigate the penetration/filtration processes.

6. Is the description of experiments and calculations sufficiently complete and precise to allow their reproduction by fellow scientists (traceability of results)? Yes.

7. Do the authors give proper credit to related work and clearly indicate their own new/original contribution? Important papers on particle imaging (examples given in item 2) are missing

8. Does the title clearly reflect the contents of the paper? Yes.

9. Does the abstract provide a concise and complete summary? Summary of the paper seems as an introduction. It needs to be rewritten to explain the summary of the work rather than the importance / introduction of the work.

10. Is the overall presentation well structured and clear? The paper is well structured and well presented.

11. Is the language fluent and precise? The language is clear and fluent.

12. Are mathematical formulae, symbols, abbreviations, and units correctly defined and used? Symbols are fine. No equation is provided.

13. Should any parts of the paper (text, formulae, figures, tables) be clarified, reduced,

combined, or eliminated? Figures 9, 10 and 11 can be shown in a single figure with three subfigures.

14. Are the number and quality of references appropriate? Fine.

15. Is the amount and quality of supplementary material appropriate? Some more information on image processing and experimental analysis should be provided in the SI.